# Perspectives and Experiences of Cardiac Rehabilitation after Stroke—A Qualitative Study

**DOI:** 10.3390/healthcare10081579

**Published:** 2022-08-19

**Authors:** Olive Lennon, Alexandra Crystal, Michelle Kwan, Caoimhe Tierney, Anne Gallagher, Sean Murphy

**Affiliations:** 1School of Public Health, Physiotherapy and Sports Science, University College Dublin, D04 V1W8 Dublin, Ireland; 2Heart House, Mater Misericordiae University Hospital, D07 KH4C Dublin, Ireland; 3Stroke Services, Mater Misericordiae University Hospital, D07 R2WY Dublin, Ireland

**Keywords:** cardiac rehabilitation, stroke, secondary prevention, qualitative

## Abstract

Cardiac rehabilitation (CR) after stroke has been proven to be a safe and feasible secondary prevention intervention. Limited qualitative data capture people’s experiences and perceptions of attending CR following stroke, but with none addressing translational aspects when CR is delivered as routine clinical care. Using a phenomenological, qualitative approach, four semi-structured focus groups were conducted with 15 individuals (60% male) who had completed CR during their stroke care pathway. Our inductive thematic analysis identified five themes. The first centred on recognising stroke as a cardiovascular disease and the applicability of CR post-stroke. The second addressed how peer understanding, camaraderie, and medical supervision created a safe and supportive environment. The third identified how the programme-built confidence supported longer-term healthy lifestyle choices in physical activity, diet, and smoking. The penultimate theme addressed the period from hospital discharge to attending CR as a time of uncertainty where many participants experienced cognitive difficulties, mood disturbances, and mental fatigue without adequate support. Lastly, participants identified unmet needs in their care pathway that included a lack of information about their referral to CR, the programme content, and accessing local supports ahead of CR. Ongoing and unmet needs both during and after CR related to self-management of secondary prevention medications, neurological issues, post-stroke fatigue, and the lack of structured support following CR completion.

## 1. Introduction

Globally, approximately 10.3 million strokes occur annually [1], which carry a continuous excess risk of death due to cardiovascular diseases for survivors [2]. Recurrent cardiovascular event rates are high at 11%, 26%, and 39% at one, five, and ten years post-stroke, respectively [3], with pharmacotherapy being the mainstay of secondary prevention [4]. Modelling data suggest that up to 80% of recurrent events could be avoided with the inclusion of exercise and dietary changes to existing secondary prevention strategies [5]. Stroke secondary prevention guidelines [6,7] reflect this with lifestyle recommendations targeting populational attributable risk factors for stroke that include physical inactivity, smoking, unsafe alcohol consumption, unhealthy diet, depression, and psychosocial stress [8]. However, an ESO/SAFE survey reported that less than half of the countries surveyed routinely provide lifestyle management after stroke, with only smoking cessation programmes being consistently available [9].

Guidelines further recommend early engagement in exercise after stroke, with the aim of improving aerobic fitness and ongoing physical activity participation [6,10,11]. Reports across multiple jurisdictions identify that rehabilitation delivered after stroke is of low aerobic intensity and associated with a high degree of sedentary time [12,13,14,15,16] and that, regardless of time since stroke, survivors have less than half the daily step count of healthy counterparts and spend the majority (>78%) of time in sedentary behaviours [17,18]. Barriers to adequate aerobic training provision during routine stroke rehabilitation include lack of equipment, time, and staff; and insufficient knowledge and skills in safe aerobic exercise prescription and implementation, compounded by a low prioritisation of aerobic exercise during routine rehabilitation [12]. Exercise and lifestyle-based interventions modelled on cardiac rehabilitation (CR) have been proposed for physical conditioning to improve habitual PA for survivors of stroke and to reduce lifestyle risk factors after stroke [19]. CR is a proven and established secondary prevention programme in coronary heart disease [20,21], and results from across Europe and North America suggest that CR is not only feasible to provide after stroke, but it is a safe, effective, and integrated approach for risk-factor reduction and to improve cardiorespiratory fitness [4,22,23,24,25]. Notably, stroke participants are reported to make similar improvements to cardiac patients during CR in cardiorespiratory fitness; risk-factor reduction; and physiological markers, including lipid profiles and blood pressure [24].

Despite promising findings for CR as a stroke secondary prevention programme, suboptimal (55–57%) uptake rates and dropout rates of more than double that of cardiac counterparts are reported for people with stroke when referred to cardiac rehabilitation [24,25]. Higher uptake rates (71%) are reported in individuals who already attend outpatient stroke rehabilitation [26,27], where the barriers recorded included lack of interest and transportation issues [26]. CR health professionals with experience in providing CR after stroke provide alternative insights, citing concerns about cognition and program engagement in participants with stroke that warrant greater consideration [24]. During a focus-group discussion, they noted that, during CR, individuals with stroke were more likely to miss appointments and engage less with risk-factor education in comparison to the cardiac participants [24].

While it is clear that CR may benefit stroke secondary prevention, it is not clear why individuals attending cardiac rehabilitation do not engage as well with risk-factor-reduction education and why those already attending stroke services are more likely to participate in CR. The experiences of individuals after stroke who participated in cardiac rehabilitation are reported in two qualitative studies associated with clinical trials of CR [25,28]. While some feasibility aspects of the CR intervention were addressed [25], results failed to provide insights into the educational and support components of the programme and could not address participants’ experiences of being referred to a cardiac rehabilitation programme during their routine stroke care pathway. Therefore, the aims of this study are translational in nature and explore, using moderated focus-group discussions, stroke participants’ perceptions of referral to and participation in cardiac rehabilitation following discharge from stroke services during routine stroke care.

## 2. Materials and Methods

### 2.1. Ethics

This research study was approved by the Mater Misericordiae University Hospital (MMUH) and Mater Private Hospital Institutional Review Board (IRB) (IRB Reference: 1/378/2006).

### 2.2. Research Team and Reflexivity

The research team comprised a primary investigator (PI) based in academia, with prior experience in conducting qualitative focus groups and stroke secondary prevention research (N = 1 female); early-stage researchers (N = 3 female) undergoing mentoring in research methods; and clinical leads from cardiac rehabilitation (N = 1 female) and stroke services (N = 1 male). The study PI, who conducted the focus-group discussions, was an independent researcher and was not linked to either the cardiac rehabilitation or stroke clinical services being discussed.

### 2.3. Study Design

A qualitative approach was utilised, underpinned by the theoretical framework of phenomenology and adhering to published guidelines [29,30]. Semi-structured focus-group discussions were conducted to explore participants’ individual and shared experiences of attending a cardiac rehabilitation programme following discharge from active stroke services. The decision to use focus groups was based on their ability to allow non-verbal communication to be observed, encourage interaction amongst participants, and highlight agreement or disagreement within the group [31,32]. They have been successfully used with individuals who have experienced stoke previously, including those with communication difficulties [33,34].

### 2.4. Participant Recruitment

Purposive sampling was used to recruit participants who were referred, in the 4 years prior to the study commencing, to an outpatient cardiac rehabilitation programme at the MMUH following hospital discharge with a diagnosis of TIA or stroke. The stroke service refers individuals with TIA and stroke to cardiac rehabilitation at their follow-up out-patient clinic when the individual is discharged from all acute stroke/rehabilitation services. This cardiac rehabilitation programme is a nurse-led, free-of-charge, outpatient, phase-three cardiac rehabilitation programme. Prior to attending the service, all participants undergo a treadmill-based exercise stress test and routine blood tests. Participants attend the programme twice a week for 8 weeks, and each session includes an hour of telemetry monitored progressive exercise training and one session per week includes a second hour-long interactive information and education session supported by an MDT, including nursing, psychology, physiotherapy, pharmacy, and dietetics. Topics covered in these interactive sessions include cardiovascular disease and its risk factors, healthy eating, managing stress, secondary prevention medications, and physical activity participation. Ten people are included on each CR course run. Study participants attended inclusive classes that comprised participants with cardiac conditions, as well as individuals with stroke.

All potential study participants received a letter by post that included a study information leaflet, potential times for focus-group discussions, and a consent form to be signed prior to participation. The contact details of the PI were listed on the information leaflet, and individuals made contact with the PI if interested in participating.

All participants were screened by phone to ensure they were over the age of 18, with sufficient cognitive ability and understanding to provide consent and sufficient comprehension of the English language to participate in the focus-group discussions. Individuals with an acute illness at the time of the focus group were excluded.

### 2.5. Setting

In keeping with recommendations for conducting focus groups with participants of older age or with disability, participants were made familiar with the setting and transportation requirements [35]. The focus-group discussions were conducted at the hospital campus where participants had previously attended for their stroke care and follow-up medical appointments and in a neutral location (i.e., a board room). Refreshments were provided. The focus groups comprised mixed-sex groups of individuals with stroke who had attended cardiac rehabilitation, the moderator (OL), and a second moderator (AC, MK, or CT) only. No third party was present during the focus-group discussions.

### 2.6. Data Collection

The focus groups were moderated by an experienced group facilitator (OL). The audio-recorded group discussions were guided by using a semi-structured question schedule (Table 1) and lasted between one and one and a half hours. The question schedule development, as guided by Krueger, integrated introductory, transition, key, and ending questions to guide the conversation and naturally flow between topics of interest [36]. Questions were generated through discussion amongst the researchers (OL, AC, MK, and CT) with respect to knowledge gaps identified in the literature and through subsequent discussion with clinical leads from cardiac rehabilitation and stroke services (AG and SM) with respect to their clinically relevant unanswered questions. The key questions with follow-on prompts enabled participants freedom to express themselves openly. A second moderator was present at each focus group (one of three early-stage researchers) and again this individual was unrelated to stroke or cardiac rehabilitation services. They composed field notes that included non-verbal cues and topics that generated group consensus or emotive responses and provided a summary of key points from the discussion at the conclusion of the focus group. Participants were asked whether they agreed that the summary provided was an accurate representation, and further comments were encouraged.

### 2.7. Data Analysis

Audio recordings were transcribed verbatim and anonymised. Reflexive thematic analysis was carried out to identify the perspectives and experiences of the cardiac-rehabilitation participants, guided by Braun and Clarke [37,38]. A reflexive, inductive approach to thematic analysis was used where, in the first stage, three researchers (AC, MK, and CT), independently of each other, immersed themselves in the data. In the next stage, the researchers each identified meaning units in the full transcripts and structured observations, initially staying close to the participants’ terminology. They reached consensus on the meaning units identified in the data through discussion. In the next stage, these meaning units were converged or split to shared meaning construct themes. Themes were then discussed and agreed upon by the three researchers. In the final stage, the three researchers (AC, MK, and CT) and the PI (OL) reviewed all themes and analytical decisions and checked and refined them against the raw data, discussing any potential biases in the identification of these themes.

## 3. Results

A total of 90 individuals with TIA/stroke were identified as having attended the cardiac rehabilitation programme in the previous four years and were contacted in writing with respect to the focus group study. A total of 22 individuals volunteered to participate, seven of whom were subsequently excluded due to scheduling conflicts (N = 5) and acute illness (N = 2). Fifteen individuals were ultimately included in the study: nine males and six females. A total of four focus groups were conducted with between three and five participants per group. A profile of participants is presented in Table 2. On average, study participants had spent 22 (±23) days in hospital at the time of their stroke and commenced cardiac rehabilitation on average 5 (±2) months following hospital discharge.

Following discussion between the researchers who independently read and coded the data initially, 22 meaning units were agreed by consensus. Discussions primarily centred on differences between researchers in their interpretation of meaning units. For example, some units were merged (e.g., the medical model and pharmacological management as discussions about the medical model focussed on medication). In contrast, it was agreed that, for example, units related to mood/mindset needed to be split. In the next stage, these agreed meaning units were converged to shared meaning construct themes by reviewers, and, again, following discussion and agreement by the three researchers and PI and further reviewing of these themes in the manuscripts, the key themes that arose following analysis of the data were agreed upon, as presented below.

### 3.1. Theme: Stroke as a Cardiovascular Disease: “We’re All in the One Boat”

The first theme identified addressed stroke as a cardiovascular disease. There was universal recognition by the focus group participants that stroke was indeed a cardiovascular disease, although a number of participants commented on the name “Cardiac Rehabilitation” as not specific enough to stroke under the cardiovascular heading.

FG4 P3: *There’s no difference really. The heart as opposed to the stroke … is all connected to the same thing.*

FG3 P2: *I think the name would put you off more than … Stupidly yeah … Well I know I was going “what the hell am I going for cardiac rehab for when I have a stroke”.*

Based on recognition of stroke as a cardiovascular disease, there was general agreement amongst participants that a cardiac rehabilitation programme was suitable for individuals with stroke, as well as those with cardiac issues:

FG1 P1: *Stroke, heart attacks, it’s the whole lot. It’s one problem. [P3over-talking with general agreement] P3: the same program is’nit?*

FG3 P2: *just the whole help -the rehab thing applies to both- the pharmacy, the exercise all of that … so I think they had a different end result to us, but ultimately the steps are the same.*

The participants similarly discussed that benefits from participation in cardiac rehabilitation were universal, irrespective of the cardiovascular disease diagnosis:

FG1 P4: *I think it did strengthen my muscles a lot … when I came here I was very weak … But like, with exercise and the talk … it was very good. [going on to say following a pause] ‘Cause some people had all sorts of different things … and so it was good for them. By the time we left everyone was feeling a lot better and more confident.*

All groups identified that they had fears and goals in common with their cardiac counterparts in relation to their condition:

FG1 P4: *I think it’s the same thing, except just different context for different complaints, do you know what I mean? The general things were … you were afraid, you were all nervous, the whole thing are we going to get better, are we going to be able to do the normal things we used to do?*

However, some participants articulated differences between themselves and individuals with cardiac issues, mostly seeing themselves as healthier:

FG1 P2: *I felt that their problems were more serious than mine.*

FG3 P2: *most of them had bypasses so they had fairly bright red scars down their chest and their leg and arm … I didn’t. So yeah I did kind of feel seriously advantaged to the cardiac patients [with P1 adding in] P1: you always feel a bit better than them … they look sicker to you, don’t they?*

In contrast to the general perception that cardiac conditions were more serious, one participant did note that stroke-related impairments meant he/she may not perform as well with physical activities:

FG4 P3: *I found a difference was the movement of limbs and stuff like that … that end of it … was from a stroke point of view … they could possibly do a wee bit more just from the physio end of it.*

### 3.2. Theme: Safe and Supported: “It’s a Safe House for You”

Participants commented on how cardiac rehabilitation provided them with a sense of security and the routine medical monitoring allowed them to feel safe when exercising at prescribed intensity levels:

FG1 P4: *I found everything about this was safe for me, and it was done good*.

FG2, P2: *You went down there every Monday, the blood pressure was taken, the heart rate … everything you know? So, you knew you were kinda … okay. That’s the difference. If you join a gym yourself, you’d say I’m gonna drop dead.*

Cardiac rehabilitation was further identified as a space where people could express their concerns freely, and the peer-led camaraderie was highlighted. Strong group agreement was recorded in the moderator’s notes during these discussion points:

FG1, P4: *It was a safe place, you could say what you like, you could talk about anything, if you had any problems, if you were worrying about something.*

FG3 P2: *There’s a great bit of banter and I think one person eggs on another without pushing anyone beyond their limits … There was great camaraderie and people were egging each other on and willing each other to be better.*

The feeling that you were understood by others was picked out as promoting a supportive environment that encompassed both peers and programme staff:

FG4 P2: *… (you) realise that you are not on your own here.*

FG2 P5: *You feel like they understand where you are coming from.*

FG4 P3: *And she knew when to push people. And the person needed that push. I thought they were very good … they knew when to and when not to.*

### 3.3. Theme: Building Confidence and Affecting Change: “It Changed Your Mindset, but It also Made You Get Up and Do”

Participants identified building confidence as an important and often undervalued aspect of their recovery after stroke:

FG1 P4: *but you need to have the confidence back. And I think they probably don’t express how important it is to get back … yourself. You know, you’ve lost a bit of yourself in confidence, so you need to get it back before you start living again. They don’t know the seriousness of that. Or how necessary it is, or how helpful it is …*


This participant elaborated on how the cardiac rehabilitation programme had allowed him/her to build this confidence again following their stroke.

FG1 P4: *Yeah, it really got me, like, moving again, because I was being extra careful, and doing very little, so … the kids weren’t letting me do this and that... and I came down here on my own, and I did it, and it was great getting your confidence back up, because it was a bit scary … you know …*


A number of participants across groups also commented on how family members had restricted their activities following hospital discharge after their stroke and how cardiac rehabilitation enabled them to break this cycle:

FG4 P2: *like you weren’t going to overstep the mark down at rehab. You knew how far you could go, at home everything you were doing was being watched … … what I found was that people see you as sick and people would then try and control you. That was one of my problems I couldn’t handle people controlling me the way that they were. Yah … it was a big battle for me. Up until I started to go [to cardiac rehabilitation].*

Confidence building in longer-term self-management activities was also described by participants:

FG2 P3: *it gave me the confidence actually to then to … at the end of it I actually joined a gym. Only because of the confidence do you get me? Not just on that but it explained the medication to me, explained the diet, explained the psychological things as well.*

Participants acknowledged the importance of the educational and information provision components of the cardiac rehabilitation programme, noting in the main that the information they received was well delivered and comprehensive for their condition:

FG2 P1: *It was good and very thorough like, they didn’t only focus on the physical they did the nutrition and the everything. [with FG2, P3 adding] P3: They talked in my language.*

FG3 P2: *They were great they were … they were super at their delivery I think no matter who you were …*


Conversely, one man identified that the education provided was too medicalised and too long:

FG1 P3: *But I … think with education … that with some of the talks here … (I) shut down after 15–20 min with these talks … because there’s too much up there … you know … it’s not basic talk … to my mind basic talk. No … medical terms that I would never know the meaning of …*


Participants identified a cognitive shift towards a more proactive approach to healthy living following the programme:

FG3 P2: *I think I just put on different glasses after the, after the programme was finished.*

Adopting healthier lifestyles as a result of engagement in cardiac rehabilitation and their increased awareness of risk reduction were also addressed by the focus-group participants. The aspects of healthy lifestyle that were addressed included the following:Exercise participation:

FG1 P4: *I was encouraged to walk and things like that. And, you know the DCU thing (community exercise scheme) might not have arisen if I hadn’t come here … and I did that for a while and I’m sure that was good for me … [With P1 responding] P1: Well I know … you’ve sparked off a thought for me … If I hadn’t been here I wouldn’t have known that it was important to exercise … and uh … simple as that … I don’t know now … I don’t know if other people were like that?*


Smoking cessation:


FG4 P2: *And it wasn’t easy, and I had to give up cigarettes and all and it wasn’t easy, but I came out the other end of it, I’m really grateful for everybody.*


Dietary changes and awareness:


FG3, P2: *I am more conscious of what I eat.*

FG4 P3: *I learned simple things … like lunches going to work now are pasta and chicken or salad and chicken with tomatoes with boiled eggs. Usually at lunch times I’d say ah sure I’ll go grab a burger or a roll or something … so from that end of it a lot of that has moved on.*

FG2 P1: *Ya. They taught us about sugars that you might not know … [with P3 talking over] P3: And salt and all that … [P1 agreeing] P1: Ya. Ya. And how to read labels [with P3 agreeing] P3: Ya … I still check the salt when I’m going out. [with P2 joining in] P2: Ya that’s it. Me too still. Something I would’ve never thought of.*

### 3.4. Theme: Psychological Issues and Mental Fatigue: “Messed up Stuff in Your Head”

Participants discussed openly how they experienced many mental health issues after stroke, notably in the period of time between discharge from stroke rehabilitation services and attending the cardiac rehabilitation programme. When the participants were asked why they thought the uptake of cardiac rehabilitation after stroke was low, they often reflected on their own mood at the time of referral, and many lengthy discussions stemmed from this prompt. Feeling down or depressed following hospital discharge was frequently discussed amongst participants.

FG2 P3: *I went through a bad phase of being quite down*

FG4 P2: *ehm but at that time I would have just gone into a depression I know, I still do … [With P3 talking over emphatically] P3: I hit depression. I wasn’t diagnosed, I didn’t go to the doctor. I knew I had it. My wife has it, so I know exactly what all the signs are.*

Anxiety issues were similarly identified by a number of participants following their stroke, and the anxiety was often associated with fear related to their illness.

FG4 P2: *The only thing I find in my mind is racing all the time … I’m always, it never sleeps. It’s always on the go all the time.*

FG1 P5: *I don’t know, afraid, maybe … Didn’t know if I was gonna die. [Participant 4 nodding as well and moderator asks: Were you afraid too?] P4: Yeah, yeah … As I said to you, I was a bit nervous, and they wouldn’t let me do anything … and … I seemed to be losing myself at home so I thought...*

Heightened irritability and the effect that this had on family were discussed in detail by one participant:

FG4 P3: *your personality is changed … you were snappy where you weren’t before so that’s a massive change for everyone who lives around you.*

Cognitive difficulties were identified by many participants in the focus groups. A primary issue discussed was difficulty with memory:

FG4 P3: *it was my memory was bunched at the time.*

FG2 P2: *For me it was memory and that.*

However, other cognitive-processing difficulties were discussed by participants:

FG2 P5: *But I was at a very low time. I mean I remember trying to make the bed and I was just … It was just beyond me to put a duvet cover on. Something I would’ve done in minutes*

FG4 P3: *Like I’m talking like a five-year-old kids’ jigsaw and I would look at it and my head went down to the floor … like I can’t even do a five-year-old’s jigsaw.*

Some participants, in reflecting on their own issues, speculated that these cognitive difficulties may contribute to low programme uptake after stroke:

FG2 P5: [on referral to cardiac rehabilitation] *I think that they’re totally confused* [P3 agreeing] P3: *I think so, ya*. [P5 continues] P5: *There can be an awful lot going on*. [P3 agreeing] P3: *Ya, ya. Your body is in shock after what’s happened to you. Not just your body but your mind and all, you know? … And It’s really hard to explain to people how you feel … … what’s in your head is not what’s coming out* [with P5 agreeing strongly]

Post-stroke mental fatigue was a topic frequently raised and was discussed in all focus groups by participants, with field notes detailing that the effects of mental fatigue after stroke were a particularly emotive topic, as exemplified in the following extracts.

FG4 P1: *yeah … Like it’s just you just feel like your whole head is just going to close down and that’s just it*

FG3 P2: *… fatigue, I know for me … like it was hugely debilitating … ehm and with that … the whole … you know the whole sensory overload … Ehm I had a huge issue with fatigue after my stroke, ehm … … . (Long pause) ehm … I just I … … (P2 gets very emotional) it’s almost paralysing. Sorry … ahm no its like I know I was trying to push myself … … I can’t believe I’m crying … there’s times there’s nowhere to turn as nobody knows what you’re talking about. Ehm and I think that’s why I was trying to prove that I’m not so wiped out … I slept for eighteen hours a day for the best part of two years and all this messed up stuff in your head like, you know I … I was … it was like ehm … I would describe it as bees in my head swarming and you know you’d be in a situation and I just thought there was nowhere …*


Participants talked about how being in cardiac rehabilitation helped their mental wellbeing in a number of different ways.

FG 4 P1: *I enjoyed it now to be honest with you … because it was company along with everything else and I was coming out of a dark place … you know what I’m on about?* [P2 nodding in agreement]

Participants in all focus groups picked up on stress-management skills as important, and the mindfulness component of the cardiac rehabilitation programme was discussed as being very helpful in two of the groups. Participants discussed how they took action to address stress though mindfulness once the link to stroke risk was made clear and commented on the usefulness of mindfulness skills in everyday life:

FG3 P3: *I wasn’t thinking about mindfulness to do with that* [Stroke risk] *as well.* [The Cardiac rehabilitation coordinator] *… said it to us*. [P5 talking over] P5: *That’s right* [P3 continues] P3: *And I had done that* [mindfulness course] *after I had a dead link. You know?*

FG4P2: *I found the mindfulness absolutely brilliant. And I would say that got me through a lot too.*

A greater understanding of cognitive issues after stroke was attributed to participation in the cardiac rehabilitation programme by some participants:

FG4 P3: *I found which I picked up through the rehab by talking to two or three different people who were down there … where the likes of memory loss … snippets of this ehm personality change …* [P1 agreeing] P1: *… yeah … yeah*.

Improvement in memory was mentioned by one participant:

FG4 P2: *I can’t imagine what it would have been like if I hadn’t been [to cardiac rehabilitation], even though now I’m only saying I’m beginning to come out of (a) cloud but … I’m beginning to think straight again but … remembering things again.*

Better management of fatigue and the importance of pacing was further highlighted by several participants in the groups:

FG3 P2: *my fatigue was my big issue. I mean I used to … I don’t know … try to push myself that I wasn’t fatigued … but it took me long after rehab … but the girls in rehab over and over and over again were like slow down and listen to your body … anyway it eventually clicked it.*

### 3.5. Theme: Unmet Needs: “Would They Not Give More Information?”

Participants almost unanimously identified that the time between hospital discharge and commencement of the cardiac rehabilitation programme was a time of uncertainty. This lack of continuity in care was identified as stressful, with many participants feeling alone at this time, with unanswered questions:

FG4 P1: *Yeah because you know you feel you are battling on your own a bit after*

FG4 P3: *I was questioning people as I didn’t know where to go or what to do. On the internet … there was nothing out there...*

FG2 P5: *I was so confused when I did get home, a good few weeks that no one was explaining anything properly to me.*

Participants identified that, with greater information and/or advice, they could have used the time after hospital discharge more effectively:

FG4 P1: *I think maybe even being told that in the hospital like maybe give yourself five minutes’ walk today and maybe do that for a couple of days and then maybe do ten minutes … this is right after your operation or after your stroke or whatever and then you’re kind of building yourself up anyway before.*

In the main, participants were unclear about their referral to the cardiac rehabilitation programme or the rationale for it:

FG1 P3: [re-referral] *It was … I think it was from the hospital here … Referred me … . I can’t remember who it was necessarily.*

FG3 P2: *I kind of didn’t overly understand why I was going but at the same time I was absolutely going to do whatever, whatever that would make me feel better or, yeah smooth the recovery.*

Participants identified a need for more information about the programme in advance of attending:

FG1 P4: *Would they not give more information out of hospitals, you know when patients are leaving, you know? Giving the information and explaining how good it (cardiac rehabilitation) is for them … they didn’t give any information … … there should be more talk about it … more detail about it before you leave.*

FG2 P5: *We didn’t know what to expect …*


Participants further expressed a lack of continuity in care following completion of the CR programme, identifying ongoing and unmet needs in relation to information and ongoing support:

FG2 P1: *I suppose, maybe, like with the course and things, if they could tell you about those supports or inform people … Or where you can go for them (supports). Oftentimes people need them, they just don’t know how to go about finding them afterwards.*

FG1 P3: *I’d like it to go on, not just the 6 weeks or 10 weeks or however long it might be. That there could be a again talking about myself come back once a month, do a bit of exercise.*

Several stroke participants reflected that some components of the programme could have been more specific and relevant to their needs post-stroke. Neurological consequences of stroke were reported to be a missing component in the educational aspects of the cardiac rehabilitation programme:

FG3 P2: *If they were to do a future cardiac rehab with stroke patients if there was even a little session … for neuro … (a) session for those people.*

In particular, participants noted that more information and strategies to manage post-stroke fatigue were needed:

FG4 P3: *Now I think that (fatigue) could be revisited. What I picked up about fatigue was listen to your body.*

Post-stroke secondary prevention medication regimens came up in all groups as a topic of continued uncertainty and were noted in field notes to generate a lot of anger and/or frustration:

FG1 P3: *Maybe go back and revisit one or two things. Like the tablet end of things … what I’m taking … what I’m taking every day … what is each one for.*

FG1 P2: *I remember asking at one of those about side effects of some of the tablets which I felt I was feeling and I was told no you couldn’t have that, and this was every one of the bloody tablets I was taking..And I just thought these people are supposed to be helping you with rehab.*

FG4 P3: *Like sometimes I’d be saying to myself like … jeeze will I have to take these things for the rest of my life. No, I have never questioned to ask, I just take them shove them down me neck say nothing … move on next … you know?*

Some participants called for a more holistic approach beyond pharmacotherapy for secondary prevention and were critical of a very medicalised model:

FG1 P1: *The idea is to get better, not to focus on what was wrong with you … a number of years ago. And you know they tell you it’s to … it’s to help you … you know … “we want to make sure this doesn’t happen to you again” … I don’t want it to happen to me again, but there’s a life out there somewhat, you know, and it isn’t focused on tablets … it shouldn’t be focused on tablets … You know, it should be focused on one’s wellbeing and um, a bit of positivity. Even if the doctors could be helpful a bit on that.*

## 4. Discussion

In an inclusive and mixed context where both cardiac and stroke participants attend sessions together, stroke participants reported that they felt understood and supported in CR and that staff were well equipped to monitor and help people with stroke because of similar risk factors and needs in both conditions. Education and information-provision sessions were deemed very relevant and were actively delivered in line with strategies identified as effective in the Cochrane review of information provision to individuals after stroke and their caregivers [39]. The overall CR attributes highlighted by stroke participants align with those identified in the Behaviour Change Technique Taxonomy (v1), including the technique clusters of Social Support (general and emotional) and Shaping Knowledge and to specific techniques of Monitoring and Credible Sources [40]. However, as previously reported [28], this study found that participants with stroke generally saw themselves as healthier than their cardiac counterparts (e.g., those post by-pass surgery), suggesting an inaccurate perception that stroke is a cardiovascular event with less associated ongoing cardiovascular risk. The findings further suggest that additional tailoring is required to support neurological issues and pharmacological adherence and to ensure that the health-literacy levels of materials are appropriate for all.

A relationship between health literacy and stroke education outcomes has been established in the literature, and a better understanding of this relationship in the context of secondary prevention of stroke is still required [41]. Health literacy assessment is not standard in cardiac rehabilitation programmes, and while materials were designed to be accessible to all, one participant still commented on the medicalised language used as not easily understood. Medication concerns are associated with a five-fold increase in the likelihood of nonadherence after stroke [42], and despite an interactive information session from a pharmacist during CR, many participants reported ongoing frustration with their prescribed pharmacological regimens, a finding with important implications for future CR-service-enhancement initiatives. Multi-modal behavioural interventions (that address medication compliance as a health behaviour) and self-management interventions have been shown in meta-analyses to improve compliance with prescribed secondary prevention medications after stroke [43,44]. Fatigue issues, which are highly prevalent following stroke and endure at 12 months [45,46], were clearly echoed in this study. Many participants experienced distressing fatigue symptoms and requested more support for this during CR, thereby signalling it as an important topic to address in the educational component of the programme. Similarly, while ischaemic stroke is a cardiovascular disease, it has neurological consequences that include neuromuscular weakness, spasticity, and balance and coordination issues [47], and, again, this study identified that individuals with stroke attending cardiac rehabilitation would welcome further education on these topics.

Overall, participants in this study identified that the cardiac rehabilitation programme helped them to make risk-reducing lifestyle changes in categories that included dietary changes, smoking cessation, mindfulness, and physical activity levels, similar to previous quantitative and qualitative findings where knowledge of risk-factor reduction was reinforced and/or learned in CR-based secondary prevention programmes [22,23,25,28,48]. Access to healthcare professionals for ongoing exercise and secondary prevention is a recurrent theme in the stroke literature [49,50], and cardiac rehabilitation may prove an important bridge to independent self-management of physical activity for secondary prevention after stroke. Study participants valued the comradery of exercising in a group setting, felt safe in initiating exercise, and gained a sense of confidence to exercise independently, perspectives previously reported in individuals both with stroke [4,28] and cardiac conditions [51,52,53]. The study offers further ancillary evidence of subjective improvement in physical fitness to support physical and physiological fitness parameters reported in stroke participants following CR [19,22,23,24].

A position paper on inclusive CR after stroke highlighted the need to create a brochure for patients and families by the referral source that includes eligibility criteria for CR, the referral pathway, where the programme is located, and a description and summary of benefits of CR following stroke [54]. Our study is novel in addressing participants’ understanding of CR following stroke prior to their programme attendance. Significant knowledge gaps about CR as a secondary prevention strategy after stroke were evident, with a small number of participants observing that the name CR itself is not self-explanatory in stroke. Limited knowledge about the programme content, its purpose, or, indeed, the referral made on their behalf to this service were identified. Lack of ongoing supports were further highlighted at CR programme completion, in line with reports of an absence of downstream services in the jurisdiction where this study was based [55]. Participants verbalised a need for top-up support to help sustain positive behavioural changes in the longer-term and overcome barriers to healthy-lifestyle participation, echoing previously published findings reflecting barriers to healthy lifestyle after stroke [49]. The contemporary literature does identify that many people and their families leave the hospital dissatisfied with the quality and quantity of information given during their hospital stay following a stroke [56] and highlights that transitioning from acute services to the community remains a challenge, with supports proposed to address this deficit including improved communication processes, use of transition specialists (e.g., keyworker/navigator), implementing a patient-centred discharge checklist, and establishing partnerships with community wellness programmes [57].

When participants considered why many people after stroke do not attend cardiac rehabilitation when referred, their responses, while speculative in nature, reflected on their own situation at the time of their referral. It was here that focus-group participants’ difficulties with cognitive functions, namely memory, thinking, attention, and sequencing, mainly surfaced and were discussed, along with mood problems, which were predominantly those of depression. A previous study identified that cardiac-rehabilitation professionals perceive individuals with stroke to have more cognitive difficulties and greater depressive symptoms than cardiac patients and identified that they were more likely to miss appointments and engage less with the educational components [24]. The recent national audit of stroke in Ireland [58] identified that, in the acute hospital setting, only 4% of individuals with stroke were assessed by a psychologist and 22% had their mood screened, suggesting that many of these issues may go undiagnosed. These findings again warrant consideration in future endeavours to enhance CR type models after stroke and suggest that both cognition and mood screening may be of benefit prior to programme participation.

The timing of cardiac rehabilitation after stroke varies in the published literature [4,22,23,25]. On average, participants in our study commenced their CR programme 20 weeks following their stroke or TIA and identified this intervening time between hospital discharge and cardiac rehabilitation as a period of great uncertainty. This has important implications in clinical practice where untapped potential for CR services at the time of discharge from stroke services may be missed currently. This hospital discharge window has been identified as a period where motivation for engagement is high and as a “teachable moment”, where receptiveness to education and lifestyle change exists [59,60]. Particularly in those with TIA and non-disabling stroke, this has also been identified as a time where ongoing rehabilitation and educational needs could be assessed in a more equitable way to those with moderate and severe stroke [61,62], something echoed in the findings in this study where participants identified the waiting time as a missed opportunity. Participants in this study further identified a lack of available resources and directives between hospital discharge and commencement of the cardiac rehabilitation programme. This finding is in keeping with the published literature where pressures to discharge people following stroke in a timely manner, despite complex presentations, is identified and the disconnect between acute, inpatient rehabilitation, and community services is recognised as a trigger for increased complexity [63].

The results of this study must be considered in light of contextual factors and limitations. A number of limitations exist with qualitative studies in stroke, including limited generalizability [64]. All study participants in this study completed the same cardiac rehabilitation programme, which may not be representative of all programmes and programme referral pathways or of non-attenders and of individuals who drop out of these programmes. Similarly, with the exception of one person, all study participants were of Irish ethnicity, and experiences of ethnic minority and other hard-to-reach groups in research are not represented here. Additional research capturing these experiences of cardiac rehabilitation is required. Many of the study participants had completed their cardiac rehabilitation programme some years previously, and this may have resulted in recall bias. Opinions provided by participants on the reasons for low attendance rates at cardiac rehabilitation following stroke were speculative in nature. While this is a limitation that must be considered in interpreting the findings, valuable insights into their own mindsets at the time of referral were gained that had relevance to the question. Finally, these study findings are not representative of individuals with severe disability after stroke who may have attended cardiac rehabilitation programmes.

## 5. Conclusions

Cardiac rehabilitation is a valuable and valued service for individuals with TIA and stroke. Exercise participation and information delivery in the areas of nutrition, exercise, and overall health were perceived to be effective in building confidence and promoting lifestyle modification and risk-factor reduction. However, some aspects of the service could be more specific to the needs identified by stroke participants particularly neurological aspects of stroke, including cognition issues, mood, and post-stroke fatigue. Additional supports are required during transitions in care to and from the cardiac rehabilitation service to improve programme engagement and sustain healthy lifestyle changes.

## Figures and Tables

**Table 1 healthcare-10-01579-t001:** Semi-structured focus group question schedule.

Question Type	Question	Purpose
Opening	Could you tell me when you completed this cardiac rehab program? Had you heard of cardiac rehab before your referral?	Familiarizes the person with the focus group process and introduces the major topic
Key Question 1	How did you come to be referred to cardiac rehab?	Obtain insights in a key concept.
Key Question 1 Prompts	Why do you think you were referred? Timing/feeling at the time of your referral/Influences on attendance/information	Obtain insights in a key concept. Keep conversation focused and flowing
Transition	Did you have any concerns about starting cardiac rehab?	Prepares participant to another key concept.
Key Question 2	What was your overall impression of cardiac rehab?	Obtain insights in a key concept.
Key Question 2 Prompts	What was your overall impression of the exercise component? Good/bad	Obtain insights in a key concept. Keep conversation focused and flowing
Key Question 3	What was your overall impression of the education and support?	Obtain insights in a key concept.
Key Question 3 Prompts	What did you think about the education and support provided? Good/bad/easy to understand/relevant	Obtain insights in a key concept. Keep conversation focused and flowing
Transition	What did you learn that you hadn’t known before?	Prepares participant to another key concept.
Key Question 4	What changes in your daily life to reduce your risk after stroke, if any, did you make as a result of attending cardiac rehabilitation?	Obtain insights in a key concept.
Key Question 5	How, if at all, do you think that cardiac rehab has helped you?	Obtain insights in a key concept.
Key Question 6 Prompts	How did you find the mix of cardiac and stroke cases in your group? What support did you get support from other people in the class? Things in common or not with cardiac patients/what unmet needs did you have/how did staff respond	Obtain insights in a key concept. Keep conversation focused and flowing.
Transition Question	Thinking back, what difficulties, if any, did you have with the cardiac rehab program?	Prepares participant to another key concept.
Key Question 7 Prompt Ending Question	There are many people who are referred to the cardiac rehab program but do not attend. Why do you think this is? What would make this difficult to attend for other stroke participants? If you could change one thing about the program, what would it be?	Obtain insights in a key concept.
		Allows participants to reflect on discussion and offer their position/ opinion on topics.
Ending Question	If another person with a stroke, similar to you, was referred to cardiac rehab and was wondering whether they should attend, what advice would you give them?	Allows participants to reflect on discussion and offer their position/ opinion on topics.
Summary Provided	Is this a fair and accurate representation of what we discussed today?	Bring closure to discussion and also a check.
Closing Question	Is there anything else that we should have talked about today but did not?	Allow participants to introduce areas of interest that were not discussed.

**Table 2 healthcare-10-01579-t002:** Descriptive statistics of study participants who attended cardiac rehabilitation.

Gender: *n* (%)	
Female	6 (40%)
Male	9 (60%)
Years since TIA/stroke: Median (Range)	3 (2–4)
Days spent in hospital with TIA/stroke: Mean (SD)	22.4 + 20.3
Months waiting to commence cardiac rehabilitation: Mean (SD)	5.4 + 3.0

## Data Availability

The data presented in this study are available upon request from the corresponding author.

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
