# Peer review of "Perspectives and Experiences of Cardiac Rehabilitation after Stroke—A Qualitative Study"

_healthcare, 2022, doi:10.3390/healthcare10081579_

Round 1
Reviewer 1 Report
Comments to the authors:
This manuscript entitled “Perspectives And Experiences of Cardiac Rehabilitation After 2 Stroke - A Qualitative Study”. The manuscript addressed the absence of supported transitions of care around stroke and CR and 21 highlighted that management of stroke secondary prevention medications, neurological issues, and 22 post-stroke fatigue were not fully addressed during CR. The manuscript is organized and well presented.
However some points are required to be revised
1. The aim of the work is redundant and needs to be clearly presented, with much details.
2. Line 86 “ The study PI was not involved in the design or delivery of either 86 cardiac rehabilitation or stroke clinical services. “ : I can’t understand what you mean to say.
3. In the setting section : needs more details and proper citations for the method.
4. Semi-structured focus group question schedule: who conducted these questions and why ? is there a specific base to choose these specified questions.
5. Is the attendance is able to answer the question number 8 after his experience , why?
6. Add a section for the limitation of the current study.
Author Response
Reviewer 1
We are grateful to you for reviewing our manuscript and providing guidance on how to improve it further, notably in the introduction and methods sections. We provide a detailed response to your points below.
- The aim of the work is redundant and needs tobe clearly presented, with much details.
Thank you for highlighting that we did not clearly enough present where this study addresses current knowledge-gaps. We have revised the introduction to better elucidate the aims of the study. The final 2 paragraphs in the introduction now identify the knowledge gaps that we address in this study better and highlight the translational nature of this study, as detailed below.
Despite promising findings for CR as a stroke secondary prevention programme, suboptimal (55-57%) uptake rates and dropout rates of more than double that of cardiac counterparts are reported for people with stroke when referred to cardiac rehabilitation [24,25]. Higher uptake rates (71%) are reported in individuals who already attend outpatient stroke rehabilitation [26,27], where the barriers recorded included lack of interest and transportation issues [26]. CR health professionals with experience of providing CR after stroke provide alternative insights citing concerns about cognition and program engagement that warrant investigation. [24]. They noted during a focus group discussion that individuals with stroke were more likely to miss appointments and engage less with risk factor education in comparison to the cardiac participants[24].
While it is clear that CR may benefit stroke secondary prevention, it is not clear why individuals attending cardiac rehabilitation do not engage as well with risk factor reduction education and why those already attending stroke services are more likely to participate in CR. The experiences of individuals after stroke who participated in cardiac rehabilitation are reported in two qualitative studies associated with clinical trials of CR [25,28]. While some feasibility aspects of the CR intervention were addressed [25], results failed to provide insights into the educational and support components of the programme and could not address participants’ experiences of being referred to a cardiac rehabilitation programme during their routine stroke care pathway. Therefore, the aims of this study are translational in nature and explore, using moderated focus group discussions, stroke participants’ perceptions of referral to and participation in cardiac rehabilitation following discharge from stroke services during routine stroke care.
- Line 86 “ The study PI was not involved in the design or delivery of either 86 cardiac rehabilitation or stroke clinical services. “ : I can’t understand what you mean to say.
Apologies that this was not clearly understood. The point we were making here was that the PI who conducted the study was independent of the stroke and cardiac rehabilitation services (i.e. a neutral party). This now reads as
The study PI, who conducted the focus group discussions, was an independent researcher and was not linked to either the cardiac rehabilitation or stroke clinical services being discussed.
- In the setting section : needs more details and proper citations for the method.
We were guided here in our revision by COREQ guidelines, the method cited in the study design section. We have also cited in this section a reference in relation to setting recommendations for focus groups where participants are older and/or have a disability. In the COREQ guidelines for reporting, setting is a distinct heading and requires the following to be reported under this heading: Setting of data collection: (1)Where was the data collected? e.g. home, clinic, workplace (2) Presence of nonparticipants i.e. Was anyone else present besides the participants and researchers? (3) Description of the sample including the important characteristics of the sample? e.g. demographic data. This now reads as
In keeping with recommendations for conducting focus groups with participants of older age or with disability, participants were familiar with the setting and transportation requirements [35]. The focus group discussions were conducted at the hospital campus where participants had previously attended for their stroke and follow-up medical appointments and in a neutral location (i.e. a board room). Refreshments were provided. Focus groups comprised mixed sex groups of individuals with stroke who had attended cardiac rehabilitation, the moderator (OL) and a second moderator (AC, MK, or CT) only. No third party was present during the focus group discussions.
[35]. Barrett J, Kirk S. Running focus groups with elderly and disabled elderly participants. Applied ergonomics. 2000 Dec 1;31(6):621-9.
- Semi-structured focus group question schedule: who conducted these questions and why ? is there a specific base to choose these specified questions.
We have provided more detail with respect to how these specific questions were generated. The text now reads as
Questions were generated through discussion amongst the researchers (OL, AC, MK, CT) with respect to knowledge gaps identified in the literature and through subsequent discussion with clinical leads from cardiac rehabilitation and stroke services (AG and SM) with respect to their clinically relevant, unanswered questions.
- Is the attendance is able to answer the question number 8 after his experience , why?
This question [If another person with a stroke, similar to you, was referred to cardiac rehab and was wondering whether they should attend, what advice would you give them?] was included as guided by Kruger. It was incorrectly labelled however as a key question. It was designed and has now been correctly labelled as an ending question. A question of this nature is designed as an “All things considered question” This question allows participants to reflect on the discussion and then offer their positions or opinions on topics of central importance to the researchers. It is one recommended when you are evaluating a service.
- Add a section for the limitation of the current study.
We had the limitations of the study detailed in the last paragraph. The journal template did not specify a distinct heading for this section. However in light of this comment we have rephrased this paragraph to better signal its content and included and referenced limitations of qualitative research in stroke. It now reads as
Results of this study must be considered in light of a number of contextual factors and limitations. A number of limitations exist with qualitative studies in stroke including limited generalizability [64]. All study participants in this study completed the same cardiac rehabilitation programme and may not be representative of all programmes and programme referral pathways or of non-attenders and of individuals who drop out of these programmes. Similarly, with the exception of one person, all study participants were of Irish ethnicity, and experiences of ethnic minority and other hard to reach groups in research are not represented here. Additional research capturing these experiences of cardiac rehabilitation is required. Many of the study participants had completed their cardiac rehabilitation programme some years previously which may have resulted in recall bias. Opinions provided by participants on the reasons for low attendance rates at cardiac rehabilitation following stroke were speculative in nature. While this is a limitation that must be considered in interpreting the findings, valuable insights into their own mindsets at the time of referral were gained that had relevance to the question. Finally, these study findings are not representative of individuals with severe disability after stroke who may have attended cardiac rehabilitation programmes.
Reviewer 2 Report
Cardiac rehabilitation (CR) delivered after stroke has proven to be a safe and feasible secondary prevention strategy.
Authors explore perceptions of referral to and participation in a cardiac rehabilitation service as part of a routine clinical care pathway following a stroke. Using a phe-nomenological qualitative approach with inductive thematic analysis, four semi-structured focus groups were conducted with 15 individuals with a stroke diagnosis who had completed CR. Five key themes were identified. 1) “We’re all in the same boat”, recognising common risk profiles and needs with cardiac patients and the suitability of the programme for both conditions. 2) “It’s a safehouse for you”, where medically supervised exercise and peer support aspects were especially valued. 3) “It changed your mindset, but it also made you get up and do” where enhanced self-efficacy and positive lifestyle changes were identified as a result of programme participation. 4) “Messed up stuff in your head” summarised participants shared experiences of cognitive dif-ficulties, mood disturbances and fatigue issues following their stroke and 5) “Would they not give more information” addressed the absence of supported transitions of care around stroke and CR and highlighted that management of stroke secondary prevention medications, neurological is-sues, and post-stroke fatigue were not fully addressed during CR.
The ms is interesting, however it needs improvements.
1. Abstract is not well written and does not reflect the summary of the sections
2. Methods are ok but results are badly written and difficult to follow. For example par. 3.1 is too fragmented. The reader my loose into it.
3. Is “Psychological Issues and Fatigue: “Messed up stuff in your head””” a title’
4. The same for “Unmet needs: “Would they not give more information?”
Author Response
Reviewer 2
Thank you for reviewing our manuscript and giving constructive feedback as to how it could be further improved. We address each comment individually below.
- Abstract is not well written and does not reflect the summary of the sections
We have reworded the abstract to address this point in tandem with the other reviewer’s comments. We now hope it summarises the study and results more accurately. The abstract has a 200 word limit so we have endeavored to provide sufficient detail within this constraint
Cardiac rehabilitation (CR) after stroke has proven a safe and feasible secondary prevention intervention. Limited qualitative data capture people’s experiences and perceptions of attending CR following stroke, none addressing the translational aspects when CR is delivered as routine clinical care. Using a phenomenological, qualitative approach, four semi-structured focus groups were conducted with 15 individuals (60% male) who had completed CR during their stroke care-pathway. Inductive thematic analysis identified five themes. The first centred around recognising stroke as a cardiovascular disease and the applicability of CR post-stroke. The second addressed how peer understanding, camaraderie and medical supervision created a safe and supportive environment. The third identified how the programme-built confidence supported longer-term healthy lifestyle choices in physical activity, diet and smoking. The penultimate theme addressed the period from hospital discharge to attending CR as a time of uncertainty where many participants experienced cognitive difficulties, mood disturbances and mental fatigue without adequate support. Lastly participants identified unmet needs in their care pathway that included a lack of information about their referral to CR and about the programme content and accessing local supports ahead of CR. Ongoing and unmet needs both during and after CR related to self-management of secondary prevention medications, neurological issues, post-stroke fatigue and the lack of structured support following CR completion.
- Methods are ok but results are badly written and difficult to follow. For example par. 3.1 is too fragmented. The reader my loose into it.
Thank you for this comment. We have read through the results and revised them as advised to better help the reader navigate through better. While we were following an established method for reporting and supporting with quotes, we can see that in places embedding quotes within sentences can make navigation a little difficult. We have worked through all themes to improve readability and where feasible we have merged the main sentences and followed with the relevant quotes. An example from 3.1 is provided here:
The first theme identified addressed stroke as a cardiovascular disease. There was universal recognition by the focus group participants that stroke was a cardiovascular disease, although a number of participants commented on the name “Cardiac Rehabilitation” as not specific enough to stroke under the cardiovascular heading.
FG4 P3: ’There’s no difference really. The heart as opposed to the stroke…. is all connected to the same thing.
FG3 P2: I think the name would put you off more than… Stupidly yeah… Well I know I was going ‘’what the hell am I going for cardiac rehab for when I have a stroke’’
Based on recognition of stroke as a cardiovascular disease, there was general agreement amongst participants that a cardiac rehabilitation programme was suitable for individuals with stroke as well as those with cardiac issues:
FG1 P1:. Stroke, heart attacks, it’s the whole lot. It’s one problem. [P3over-talking with general agreement] P3: the same program is’nit?
FG3 P2: just the whole help -the rehab thing applies to both- the pharmacy, the exercise all of that.. so I think they had a different end result to us, but ultimately the steps are the same.
- Is “Psychological Issues and Fatigue: “Messed up stuff in your head””” a title’
This is an identified theme from the focus groups. We note that we did not, in error, number it or the following theme as we had for previous ones. We have addressed this and also labelled all as themes more clearly for the reader
- The same for “Unmet needs: “Would they not give more information?”
As per question 3, this is now numbered and labelled clearly as a theme
Round 2
Reviewer 1 Report
The authors addresses the required comments adequately.
Some grammatical and typos errors were detected , should fix them in the final version .
Reviewer 2 Report
The authors extensively replied/answered to my comments. There are not further comments.